# Visualizing Bioabsorbable Spacer Effectiveness by Confirming the Distal-Tail of Carbon-Ion Beams: First-In-Human Report

Shintaro Shiba [1,2,*], Masahiko Okamoto [2], Makoto Sakai [3] and Tatsuya Ohno [2]

1  Department of Radiation Oncology, Shonan Kamakura General Hospital, 1370-1, Okamoto, Kamakura-City 247-8533, Kanagawa, Japan
2  Department of Radiation Oncology, Gunma University Graduate School of Medicine, 3-39-22, Showa-Machi, Maebashi-City 371-8511, Gunma, Japan
3  Gunma University Heavy Ion Medical Center, 3-39-22, Showa-Machi, Maebashi-City 371-8511, Gunma, Japan
*  Correspondence: shiba48@gunma-u.ac.jp; Tel.: +81-467-46-1717

**Abstract:** In particle therapy, bioabsorbable polyglycolic acid (PGA) spacer was developed to reduce the healthy organ irradiation dose, especially in the gastrointestinal tract. The PGA spacer is safe and effective; however, there are no reports that have confirmed whether the PGA spacer which inserted in the body actually stops the carbon-ion (C-ion) beams. Here, we visualized and confirmed that the PGA spacer stops the C-ion beams in the body based on the dose distribution using auto-activation positron emission tomography (AAPET). A 59-year-old dedifferentiated retroperitoneal liposarcoma patient underwent C-ion radiotherapy (C-ion RT) on referral. A month before C-ion RT initiation, the patient underwent PGA spacer placement. Postoperatively, the patient received 4.4 Gy (RBE) per fraction of C-ion RT, followed by AAPET. AAPET revealed lower positron emitter concentrations at the distal tissue ventral to the PGA spacer than in the planning target volume. In observing the efficacy of the PGA spacer, the AAPET images and the average count per second of the positron emitter suggested that the PGA spacer stopped the C-ion beams in the body in accordance with the dose distribution. Therefore, AAPET was useful in confirming the PGA spacer's effectiveness in this study, and the PGA spacer stopped the C-ion beams.

**Keywords:** auto-activation positron emission tomography; bioabsorbable polyglycolic acid spacer; carbon-ion radiotherapy; Bragg peak; positron emission tomography

## 1. Introduction

Carbon-ion radiotherapy (C-ion RT) has biological and physical advantages compared to photon therapy, and there are favorable clinical outcomes for various cancers [1–5]. Biologically, C-ion RT has a higher cell-killing effect due to its high linear energy transfer to radioresistant tumors, such as malignant melanoma, bone and soft tissue sarcoma, and hypoxic tumors [3–6]. Furthermore, C-ion RT has higher dose localization properties than photon therapy, including intensity-modulated and stereotactic body radiotherapies, owing to the distal tail-off due to the Bragg peak and sharp lateral penumbra, enabling high dose administration [7]. However, C-ion RT cannot be performed for tumors close to or attached to the gastrointestinal (GI) tract due to the high risk of severe toxicity except with the surgical insertion of a spacer that physically separates the tumor from the GI tract. The spacer stops the C-ion beams at the distal end or reduces the radiation dose to the lateral penumbra by keeping a distance from the GI tract [8–11]. Previously, Gore-Tex sheets were used as spacers, and a favorable clinical result for C-ion RT patients was reported [12]. However, there were infection risks and adverse effects on the GI tract due to using Gore-Tex sheets for an extended period after C-ion RT. Recently, bioabsorbable polyglycolic acid (PGA) spacer (Alfresa Pharma Corporation, Osaka, Japan) was developed in Japan, and the mainstream of spacer-placement options shifted from Gore-Tex sheets to the PGA spacer [8,10,11]. This is due to the PGA spacer's safety compared to Gore-Tex

sheets owing to the PGA spacer absorption 32 weeks after the placement. Furthermore, infection risks and adverse effects on the GI tract will be reduced. However, the PGA spacer would be absorbed to some degree during the treatment, the C-ion beam range may change unexpectedly, and the stopping power of the spacer may be reduced by carbon dioxide ($CO_2$) generated during the spacer decomposition [8]. No reports have confirmed that the inserted PGA spacer stops the C-ion beams based on the dose distribution.

Regarding dose distribution estimation, the usefulness of images owing to the detection of annihilation gamma rays (pair of 511 keV photons) using positron emission tomography (PET) and Compton camera after C-ion RT are reported [13–15]. Positron emitters are generated from nuclear interactions between incoming C-ions and target materials, whereas annihilation gamma rays are produced through the annihilation of a positron with an electron. Thus, the positron emitter distribution imaged by the PET or Compton camera correlates with the dose distribution of C-ion RT. Hence, this study visualizes and confirms that the PGA spacer stops the C-ion beams in the body in accordance with the dose distribution using auto-activation PET (AAPET).

## 2. Materials and Methods

### 2.1. Patient and Carbon-Ion Radiotherapy

A 59-year-old male Japanese patient with dedifferentiated retroperitoneal liposarcoma (clinical T2N0M0 Stage IIIA, based on the eighth edition of the Union for International Cancer Control/American Joint Committee on Cancer TNM staging system) was referred to Gunma University Heavy Ion Medical Center (GHMC) (Figure 1A–E) [16]. The patient had a liposarcoma sub nodule attached to the small intestine. A month before initiating C-ion RT, the patient underwent a sub nodule resection surgery, a PGA spacer placement for the main tumor, and a colostomy. In C-ion RT, a heavy-ion accelerator at the GHMC generated C-ion beams. Beam energies of 290 MeV/u in the vertical beam and 400 MeV/u in the horizontal beam were selected based on the tumor depth. C-ion RT doses were expressed as relative biological effectiveness (RBE) weighted dose (Gy (RBE)), defined as the physical dose multiplied by the C-ions RBE [17]. Treatment planning computed tomography (CT) images and contrast-enhanced magnetic resonance images were merged to precisely delineate the target. The gross tumor volume (GTV) and the clinical target volume (CTV) were also delineated. The CTV was obtained using a margin with an anatomical compartment of muscle or bone or at least a 5 mm margin around the GTV to include microscopic diseases. The planning target volume (PTV) included the CTV with a 3-mm margin for possible positioning errors. When the CTV overlapped with the PGA spacer, the margin was reduced accordingly. In contrast, PTV and PGA spacer overlap were acceptable, with no margin correction. When the PTV overlapped with an organ at risk (OAR), the margin was reduced accordingly. The PTV was 650 $cm^3$. The dose distribution was calculated by estimating PGA stopping power using the stopping power of water. C-ion RT was subsequently performed using layer-stacking irradiation [18]. The administered dose of C-ion RT was 70.4 Gy (RBE) in 16 fractions for four weeks (4.4 Gy (RBE) per fraction). Figure 1F presents the dose distribution of the treatment plan. Informed consent was obtained from the patient before therapy initiation, and the ethics committee of the Gunma University Graduate School of Medicine approved this study in accordance with the Declaration of Helsinki.

### 2.2. Auto-Activation Positron Emission Tomography

A clinical PET-CT scanner (Eminence STARGATE; Shimadzu Corporation, Kyoto, Japan) acquired the distribution of positron emitters after C-ion irradiation. Auto-activation PET-CT was performed in the third and fifth fractions of the 16 fractions. In this study, the beam angle was selected such that the C-ion beam stopped at the PGA spacer placed between the distal end of the tumor and the proximal end of the GI tract. This irradiation was performed using a 180° vertical beam. The irradiation time of the C-ion beams was 248 and 260 s for the third and fifth fractions, respectively. After the irradiation, the patient

was immediately taken to a PET room near the treatment room and placed in a supine position, and the PET-CT scan was started 10 min after C-ion RT. Subsequently, the PET and CT images simultaneously taken were merged using the coordinate matching technique for comparison with the dose distribution of the treatment plan. The average count per second (CPS) of the positron emitters was calculated in the following areas: PTV, the PGA spacer, the distal tissue ventral to the PGA spacer (out of the irradiation field at the distal side), and the right side of the abdomen (out of the irradiation field on the lateral side). Figure 2 presents a schematic CT image of the CPS estimation area. The CPS were compared using *t*-tests for all combinations. Statistical significance was set at $p < 0.05$. Additionally, we measured the average CT value of the spacers on the treatment planning CT and the CT images taken at the third and fifth fractions.

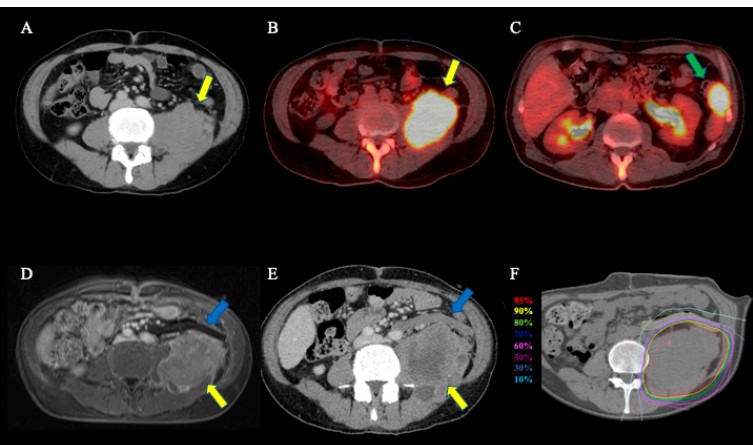

**Figure 1.** Radiological images before C-ion RT and dose distribution. (**A**) Contrast-enhanced CT. The yellow arrow reveals the tumor. (**B**) FDG-PET/CT. The yellow arrow reveals the tumor with an abnormal FDG uptake (standardized uptake value max = 32.67). (**C**) FDG-PET/CT. The green arrow presents the sub nodule with an abnormal FDG uptake (standardized uptake value max = 8.98). (**D**) CT images after the spacer placement. The yellow arrow reveals the tumor, and the blue arrow presents the spacer. (**E**) Contrast-enhanced magnetic resonance image of the spacer placement. The yellow arrow reveals the tumor, and the blue arrow shows the spacer. (**F**) Dose distribution on axial CT images. The area within the red outline is the tumor. Highlighted are the 95% (red), 90% (yellow), 80% (green), 70% (deep blue), 60% (magenta), 50% (purple), 30% (blue), and 10% (light blue) isodose curves (100% was 70.4 Gy (RBE)).

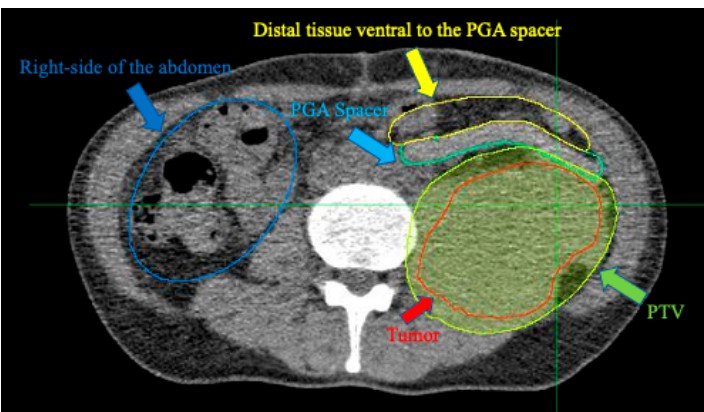

**Figure 2.** Contours on CT images used in CPS calculations. The red, light green, light blue, yellow, and blue outlines area and arrows reveal the tumor, PTV, the PGA spacer, distal tissue ventral to the PGA spacer (out of the irradiation field at the distal side), and the right side of the abdomen (out of the irradiation field on the lateral side).

## 3. Results

The patient completed C-ion RT as scheduled. Six months after initiating C-ion RT, the patient developed no toxicities. The AAPET images were merged with the CT images simultaneously taken. Figures 3A and 4A present the dose distribution with the 180° beam in the treatment planning, Figures 3B and 4B present the CT images taken simultaneously using the PET scan, Figures 3C and 4C present the AAPET images before merging, whereas Figures 3D and 4D present the AAPET images after merging with the CT images. Figures 3D and 4D reveal that the positron emitter distribution was consistent with the high-dose areas of C-ion RT and that the positron emitter concentration in the distal tissue ventral to the PGA spacer was lower than that in the PTV. The spacer volume on the treatment planning CT and the CT images taken at the third and fifth fraction did not differ (134.8 cm$^3$, 135.0 cm$^3$, and 135.1 cm$^3$, respectively); however, the mean CT Hounsfield unit (HU) values were 65.32 HU, 37.36 HU, and 29.75 HU, respectively. The overlap between the PGA spacer and the PTV was 0–3 mm, and the spacer thickness was 8–11 mm without PTV overlap. Figure 5 shows the average and standard deviations for all pixel value data for the CPS of the positron emitter in both the third and fifth fraction images. Additionally, the CPS of the positron emitter in the PTV, the PGA spacer, the distal tissue ventral to the PGA spacer (out of the irradiation field at the distal side), and the right side of the abdomen (out of the irradiation field on the lateral side) were 13.8 ± 6.9, 6.1 ± 4.1, 3.9 ± 2.6, and 1.1 ± 1.3, respectively (Figure 5). The CPS of the positron emitter in the distal tissue ventral to the PGA spacer and the right side of the abdomen presented a significantly lower concentration than those in the PTV (each *p*-value < 0.05). These AAPET images and the average CPS of the positron emitter suggested that the PGA spacer stopped the C-ion beams in the body in accordance with the dose distribution.

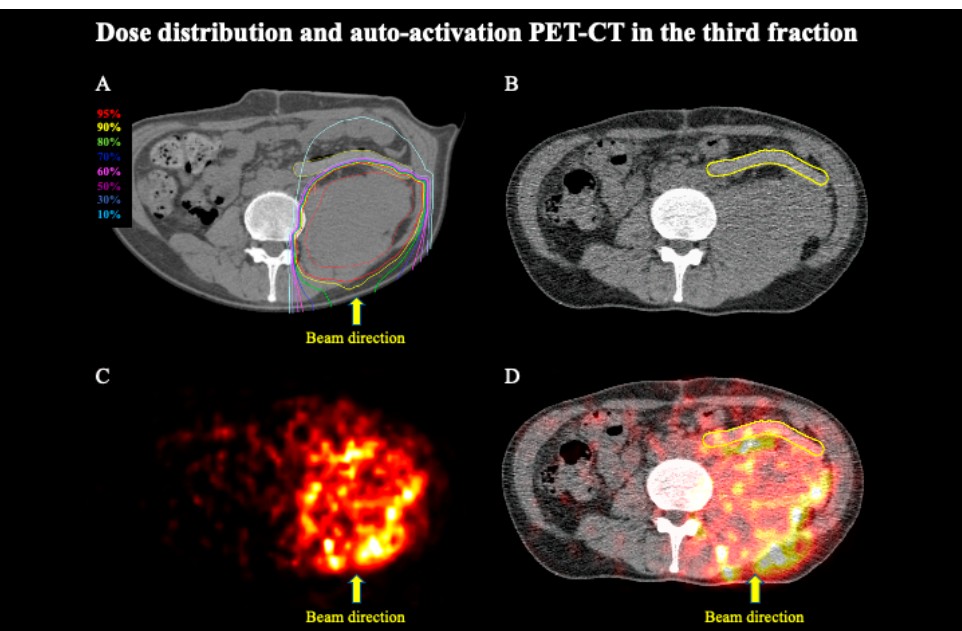

**Figure 3.** Dose distribution of C-ion RT and AAPET images with a 180° beam in the third fraction. (**A**) Dose distribution of C-ion RT with 180° beam. The area within the red outline is the tumor. Highlighted are the 95% (red), 90% (yellow), 80% (green), 70% (deep blue), 60% (magenta), 50% (purple), 30% (blue), and 10% (light blue) isodose curves (100% was 4.4 Gy (RBE)). (**B**) CT image simultaneously taken with the PET scan. The area within the yellow outline is the spacer. (**C**) AAPET image. (**D**) Positron emitter distributions obtained using merged CT and AAPET images. The area within the yellow outline is the spacer.

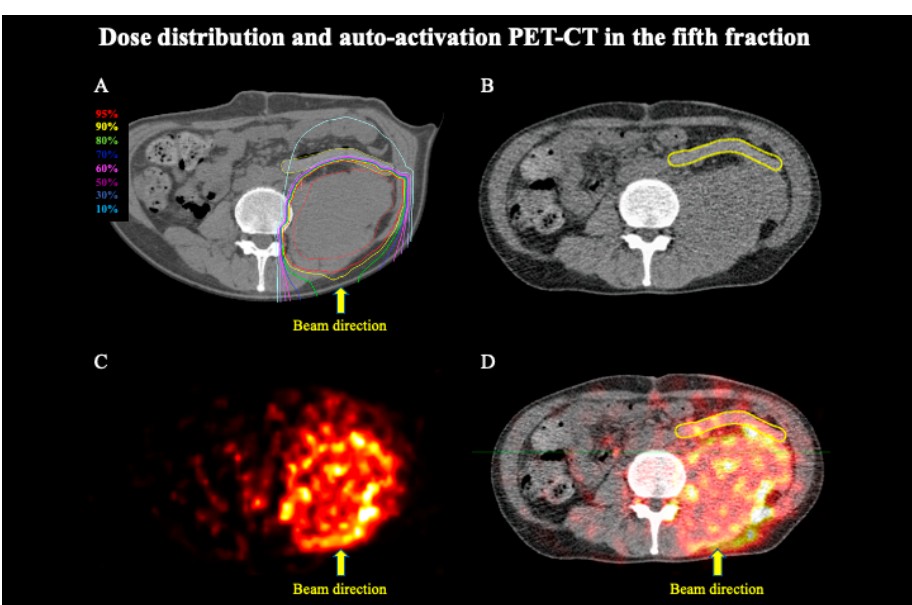

**Figure 4.** Dose distribution of C-ion RT and AAPET images with a 180° beam in the fifth fraction. (**A**) Dose distribution of C-ion RT with 180° beam. The area within the red outline is the tumor. Highlighted are the 95% (red), 90% (yellow), 80% (green), 70% (deep blue), 60% (magenta), 50% (purple), 30% (blue), and 10% (light blue) isodose curves (100% was 4.4 Gy (RBE)). (**B**) CT image simultaneously taken with the PET scan. The area within the yellow outline is the spacer. (**C**) AAPET image. (**D**) Positron emitter distributions obtained by merged CT and AAPET images. The area within the yellow outline is the spacer.

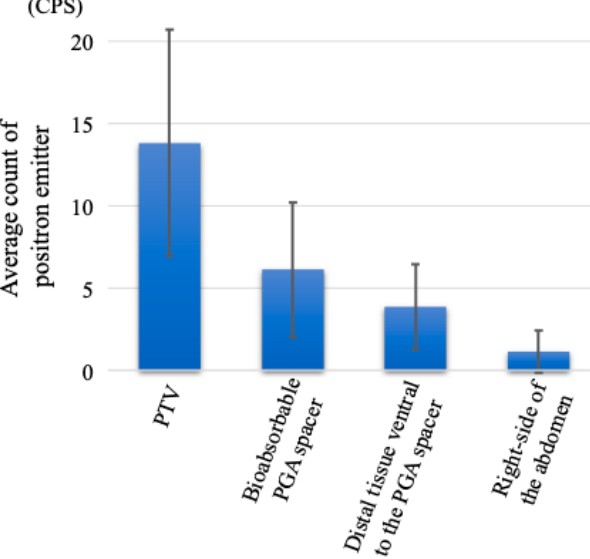

**Figure 5.** Counts per second of positron emitters in the planning target volume, the bioabsorbable PGA spacer, the distal tissue ventral to the PGA spacer, and the right side of the abdomen (out of the irradiation field). Data were presented as the mean ± standard deviation for all pixel value data in both the third and fifth fraction images for each contour. Data were compared for all combinations, and *p*-values were less than 0.05 for all tests.

## 4. Discussion

Using AAPET, we visualized and confirmed that the PGA spacer stops the C-ion beams in the body in accordance with the dose distribution. To the best of our knowledge, this is the first report to visualize and confirm the effectiveness of the bioabsorbable PGA spacer in stopping C-ion beams in the body.

The safety of the PGA spacer has been reported [8], and another report of dosimetric comparison in a simulation study revealed that the PGA spacer improves dose distribution [10]. In this case, the tumor was close to the GI tract, and the PGA spacer was inserted, resulting in a reduced C-ion RT dose for the GI tract, and the treatment was safely performed. After C-ion RT, no GI toxicities or spacer-related toxicities were observed. This safe treatment is probably due to the $\geq 8$ mm spacer thickness without PTV overlap, sufficiently reducing the C-ion dose to the distal tissue ventral to the PGA spacer. In contrast, the period between diagnosis and C-ion RT initiation was two months, and tumor growth was observed (Figure 1A,E). Notably, for an extended waiting period, especially for rapidly growing tumors, it is possible to miss the timing of the local treatment; therefore, efforts should be made to shorten the waiting period between the diagnosis, spacer insertion, and C-ion RT initiation.

The PGA spacer eventually decomposes into water and $CO_2$; therefore, the $CO_2$ gas generated by the PGA spacer decomposition might be stored in the spacer. Stopping the C-ion beams at the distal end requires materials (e.g., water) with the stopping power inside the spacer, and if the inside of the spacer is air such as $CO_2$, the C-ion beams will pass through the spacer. Furthermore, as previously reported, the CT images taken at the third and fifth irradiation compared to the treatment planning CT revealed decreased CT values of the PGA spacer [11]. However, the change in CT values was slight, and the range of the C-ion beam inside the PGA was approximately 2–3%. Moreover, no apparent void areas due to gas generation were observed within the PGA. Therefore, the C-ion beams would have stopped at almost the same location as in the treatment plan. This can be confirmed using the PET images, which reveal a significantly higher CPS in the PTV and a gradual decrease on its distal side (the PGA and GI region). On the other hand, a minimal amount of gas generation is not clinically problematic. However, if the void area due to gas generation and the CT value changes are large, the range of C-ion beams may change when the spacer is used to stop the C-ion beams at the distal end. Therefore, evaluating the size of the spacer, the void newly generated, and the CT values of the spacer are necessary during the treatment. Suppose the spacer undergoes such a change, AAPET imaging can confirm that the irradiated area is consistent with the dose distribution.

Comparing the CPS between the right side of the abdomen (out of the irradiation field on the lateral side) and the distal tissue ventral to the PGA spacer (out of the irradiation field at the distal side), the CPS in the distal tissue ventral to the PGA spacer was slightly higher than that of the right side of the abdomen. In C-ion RT, the nuclear reactions produce an excess dose at the terminal end of the Bragg peak (the fragment tail), and a previous phantom study reported that the fragment tail of C-ion beams is extended on the distal side, which could be confirmed using AAPET [19]. In this study, we confirmed the fragment tail of C-ion beams using AAPET. We considered that our CPS in the distal tissue ventral to the PGA spacer, which is higher than that on the right side of the abdomen, would express the fragment tail with a low dose of C-ion beams. The washout effect should be considered when estimating the C-ion beam range using AAPET [15,20]. However, there are no blood or lymphatic vessels within the PGA, and the washout effect should be smaller than in the PTV region.

This study had limitations. In our facility, the PET is located separately from the treatment room owing to its large size. Thus, it is impossible to monitor in real time using PET during C-ion RT irradiation. Therefore, only confirming the irradiated area after irradiation is feasible in our facility, and it will be challenging to apply AAPET to adaptive C-ion RT currently.

## 5. Conclusions

We observed the PGA spacer efficacy using AAPET, which visualized that the spacer stopped C-ion beams in the body in accordance with the dose distribution. AAPET was useful in confirming the PGA spacer's effectiveness in this study, and the PGA spacer stopped the C-ion beams, allowing for safe treatment.

**Author Contributions:** Conceptualization, S.S.; methodology, S.S. and M.S.; validation, S.S.; formal analysis, S.S.; investigation, S.S.; resources, S.S. and M.O.; data curation, S.S.; writing—original draft preparation, S.S. and M.S.; writing—review and editing, M.O. and T.O.; visualization, S.S. and M.S.; supervision, T.O.; project administration, M.O.; funding acquisition, S.S. and T.O. All authors have read and agreed to the published version of the manuscript.

**Funding:** This research received no external funding.

**Institutional Review Board Statement:** The study was conducted according to the guidelines of the Declaration of Helsinki and approved by the Institutional Review Board of Gunma University (date of approval: 17 August 2021).

**Informed Consent Statement:** Informed consent was obtained from the patient.

**Data Availability Statement:** Research data are stored in an institutional repository and will be shared upon request to the corresponding author.

**Acknowledgments:** The authors thank our colleagues at Gunma University Heavy Ion Medical Center and Department of Radiation Oncology, Gunma University Graduate School of Medicine.

**Conflicts of Interest:** T.O. received research funding from Hitachi. All other authors have no conflicts of interest to declare.

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
