# Peer review of "Visualizing Bioabsorbable Spacer Effectiveness by Confirming the Distal-Tail of Carbon-Ion Beams: First-In-Human Report"

_tomography, doi:10.3390/tomography8050195_

Round 1
Reviewer 1 Report
Carbon ion radiation therapy (CIRT), thought, is not world widely used radiation therapy for malignancies due to its relatively shorter history compared with other radiation therapies. CIRT has been studied for a variety types of malignancy and proved its effectiveness of the therapy. The authors are a reputational health team and experts in the area. Here in this manuscript the authors demonstrated their first case study on the PGA bio-absorbable spacer proper application to reduce irradiation on the nearby healthy organs, which was also validated by PET-CT right after two different irradiation fractions. Overall, the manuscript was well written and little language flaw has been found. However, I have to raise a few comments and questions on the presentation of the research after reading. Definitely, I would like to recommend it to be published in the journal of Tomography after the authors address the following concerns on the current report and study.
1. I noticed 18F-FDG was provided to the patient, and I suggest the does amounts and uptake values of the tumor and nodule should be reported as well.
2. No info of the PGA spacer has been provided. Where was it manufactured?
3. Citation should be added for the statements from line 40 to 43.
4. Could you please try to reduce some of the self citations due to the current high self-citation ratio? Might be able to find some of the research protocol settings in some more original reports?
5. In Figure 5, the data for the third fraction and fifth fraction seemed to be reported as the average number with relatively large SDs. If it's possible to show the data independently so that we can analyze the counts per second of positron emitters change over the progression. If the CPS for each localization getting bigger or just change randomly?
6. Line 185 to line 187, is one report on the safety the reference 10 and another report is reference 12? Seems a little bit confusing here.
7. Line 193, Figures, letter F should be capitalized.
8. Could you please discuss on the Bragg peak of carbon ions in the discussion section as well? I am a little bit confused on the Bragg peak versus the PGA spacer function here. Did the spacer as a material really blocked (stopped) the carbon ion beam or the space created by the spacer just located at the travel limit of the carbon ion beam because of the Bragg peak as a consequence of calculated energy control of the beam based on the size of the tumor?
9. Lastly, did the therapy effectively control the tumor progression in this case? Or did this PGA spacer surgery helped increase the life span of the patient?
Please let me know if you have any questions on my comments and concerns above.
Author Response
Carbon ion radiation therapy (CIRT), thought, is not world widely used radiation therapy for malignancies due to its relatively shorter history compared with other radiation therapies. CIRT has been studied for a variety types of malignancy and proved its effectiveness of the therapy. The authors are a reputational health team and experts in the area. Here in this manuscript the authors demonstrated their first case study on the PGA bio-absorbable spacer proper application to reduce irradiation on the nearby healthy organs, which was also validated by PET-CT right after two different irradiation fractions. Overall, the manuscript was well written and little language flaw has been found. However, I have to raise a few comments and questions on the presentation of the research after reading. Definitely, I would like to recommend it to be published in the journal of Tomography after the authors address the following concerns on the current report and study.
Response: We sincerely thank you for evaluating our manuscript and for the encouraging comments. After revising our manuscript according to your suggestions, we are resubmitting it for further review. We hope we have addressed all your concerns.
- I noticed 18F-FDG was provided to the patient, and I suggest the does amounts and uptake values of the tumor and nodule should be reported as well.
Response: Thanks for the comment. We added standardized uptake value max of the tumor and the sub nodule in the Figure 1 legend; “(B) FDG-PET/CT. The yellow arrow reveals the tumor with an abnormal FDG uptake (standard-ized uptake value max = 32.67). (C) FDG-PET/CT. The green arrow presents the sub nodule with an abnormal FDG uptake (standardized uptake value max = 8.98).”
- No info of the PGA spacer has been provided. Where was it manufactured?
Response: Thanks for the important comment. We added the company name that developed PGA spacer to the section of Introduction; “Recently, bioabsorbable polyglycolic acid (PGA) spacer (Alfresa Pharma Corporation, Osaka, Japan) was developed in Japan, and the mainstream of spacer-placement options shifted from Gore-Tex sheets to PGA spacer.
- Citation should be added for the statements from line 40 to 43.
Response: Thanks for the comment. We added the following references in the line from 40 to 43.
- Sasaki, R.; Demizu, Y.; Yamashita, T.; Komatsu, S.; Akasaka, H.; Miyawaki, D.; Yoshida, K.; Wang, T.; Okimoto, T.; Fukumoto, T. First-In-Human Phase 1 Study of a Nonwoven Fabric Bioabsorbable Spacer for Particle Therapy: Space-Making Particle Therapy (SMPT). Adv Radiat Oncol 2019, 4, 729–37.
- Lorenzo C, Andrea P, Barbara V, Denis P, Rosaria FM, Piero F, Viviana V, Alberto I, Mario C, Brugnatelli S, Tommaso D, Bugada D, Marcello M, Mario A, Francesca V, Roberto O, Paolo D. Surgical spacer placement prior carbon ion radiotherapy (CIRT): an effective feasible strategy to improve the treatment for sacral chordoma. World J Surg Oncol. 2016 Aug 9;14(1):211.
- Shiba, S.; Okamoto, M.; Tashiro, M.; Ogawa, H.; Osone, K.; Yanagawa, T.; Kohama, I.; Okazaki, S.; Miyasaka, Y.; Osu, N.; Chikuda, H.; Saeki, H.; Ohno, T. Rectal dose-sparing effect with bioabsorbable spacer placement in carbon ion radiotherapy for sacral chordoma: dosimetric comparison of a simulation study. J Radiat Res 2021, 62, 549–55.
- Serizawa, I.; Kusano, Y.; Kano, K.; Shima, S.; Tsuchida, K.; Takakusagi, Y.; Mizoguchi, N.; Kamada, T.; Yoshida, D.; Katoh, H. Three cases of retroperitoneal sarcoma in which bioabsorbable spacers (bioabsorbable polyglycolic acid spacers) were in-serted prior to carbon ion radiotherapy. J Radiat Res 2022, 63, 296–302.
- Could you please try to reduce some of the self citations due to the current high self-citation ratio? Might be able to find some of the research protocol settings in some more original reports?
Response: Thanks for the important comment. We apologize for the many self-citations. Citations have been deleted or changed to reports from other research groups as long as possible. The changes are as follows.
From Shiba S, et al. Cancers (Basel) 2021, 13, 1099 to Mohamad O, et al. Oncotarget. 2018 May 1;9(33):22976-22985.
Shiba S, et al. Anticancer Res 2020, 40, 459–64 was deleted.
From Shiba S, et al. Front Oncol 2020, 10, 635 to Ammar C, et al. Phys Med Biol. 2014 Dec 7;59(23):7229-44.
- In Figure 5, the data for the third fraction and fifth fraction seemed to be reported as the average number with relatively large SDs. If it's possible to show the data independently so that we can analyze the counts per second of positron emitters change over the progression. If the CPS for each localization getting bigger or just change randomly?
Response: Thanks for the insightful comment. We show the graphs of third fraction, fifth fraction, and the mean value of them as follows.
These SDs are due to differences in CPS for each voxel in the region of interest. And there might be randomly change not getting bigger in CPS as carbon-ion radiotherapy progressed.
Figure. Counts per second of positron emitters of third fraction (A), fifth fraction (B), and the mean value of them (C).
- Line 185 to line 187, is one report on the safety the reference 10 and another report is reference 12? Seems a little bit confusing here.
Response: Thanks for the comment. The report of confirming safety for humans is the reference 10 and the reports of the dose distribution improvement is reference 12. References have been renumbered because I have rearranged the citations (Reference number is changed from 10 to 8 and 12 to 10). We correct the sentence clarity; “The safety of the PGA spacer has been reported [8], and another report of dosimetric comparison in a simulation study revealed that the PGA spacer improves dose distribution [10].”
- Line 193, Figures, letter F should be capitalized.
Response: Thanks for the comment. We correct from “figures” to “Figures”.
- Could you please discuss on the Bragg peak of carbon ions in the discussion section as well? I am a little bit confused on the Bragg peak versus the PGA spacer function here. Did the spacer as a material really blocked (stopped) the carbon ion beam or the space created by the spacer just located at the travel limit of the carbon ion beam because of the Bragg peak as a consequence of calculated energy control of the beam based on the size of the tumor?
Response: Thanks for the comment. Carbon-ion beams have the Bragg peak, and the physical dose falls steeply on its distal side called distal tail-off and this nature can be used to create a highly concentrate dose distribution. In treatment planning, it may be possible to create a plan that can safely treat the patient if reproducibility, intra-fractional errors, and uncertainty are ignored. However, in actual treatment, daily setup errors, daily internal organ positioning, and intra-fractional errors may prevent irradiation according to the dose distribution of the treatment planning and may cause unexpected excess dose administration to the healthy organ. Additionally, although we considered that a carbon-ion dose of 64 Gy (RBE) or more is necessary to control the sarcoma, the tolerable dose for the gastrointestinal tract is 60 Gy (RBE), and when the tumor is in contact with the gastrointestinal tract, a tumor-controllable dose cannot be administered, which is not helped by the use of steep distal tail-off. Therefore, a spacer insertion that physically separates the tumor from the GI tract is useful in terms of safety and efficacy of carbon-ion radiotherapy. This is because the range of the carbon ion beams is adjusted by beam energy, range shifter, and bolus, and the spacer is necessary not because of the range of carbon-ion beams, but because of the distance to the gastrointestinal tract and the tumor. Therefore, spacer insertions are needed to the patients with tumors close to or attached to the gastrointestinal tract.
- Lastly, did the therapy effectively control the tumor progression in this case? Or did this PGA spacer surgery helped increase the life span of the patient?
Response: Thanks for the comment. Unfortunately, this patient has had a local recurrence. However, the recurrence was from the center of the tumor, and the tumor of gastrointestinal tract side where the spacer was inserted was controlled.
The patient was received re-irradiation with carbon-ion radiotherapy as a salvage treatment and is currently under follow-up. We believe that carbon-ion radiotherapy with spacer insertion might be useful and help increase the life span of the patient. This is because if the tumor on the gastrointestinal side is not controlled, chemotherapy or best supportive care are the only treatment options, in which case the prognosis is expected to be shorter.

Reviewer 2 Report
There have been a number of studies that have performed similar searches; what does your article add to literature compared to these other ones? I did not go back to literature, but how can your conclusion be different than all others in the past?
Author Response
There have been a number of studies that have performed similar searches; what does your article add to literature compared to these other ones? I did not go back to literature, but how can your conclusion be different than all others in the past?
Response: Thanks for the comment. Recently, PGA spacer was developed in Japan, and previous report revealed the safety and dose distribution improvement was confirmed. However, no report confirms that the inserted PGA spacer stops the carbon-ion beams based on the dose distribution. Therefore, we performed the study which visualizes and confirms that the PGA spacer stops the carbon-ion beams in the body in accordance with the dose distribution using auto-activation PET.
